# Gene expression analysis suggests immunosuppressive roles of endolysosomes in glioblastoma

**Michael A. Sun**[1,2,3], **Haipei Yao**[1,2,3], **Qing Yang**[4], **Christopher J. Pirozzi**[1,2], **Vidyalakshmi Chandramohan**[1,5], **David M. Ashley**[1,5], **Yiping He**[1,2]*

1 The Preston Robert Tisch Brain Tumor Center, Duke University Medical Center, Durham, NC, United States of America, 2 Department of Pathology, Duke University Medical Center, Durham, NC, United States of America, 3 Pathology Graduate Program, Duke University Medical Center, Durham, NC, United States of America, 4 Duke University School of Nursing, Durham, NC, United States of America, 5 Department of Neurosurgery, Duke University Medical Center, Durham, NC, United States of America

* yiping.he@duke.edu

**Data Availability Statement:** All relevant data are available from the GDC data portal (https://portal.gdc.cancer.gov/).

## Abstract

Targeting endolysosomes is a strategy extensively pursued for treating cancers, including glioblastomas (GBMs), on the basis that the intact function of these subcellular organelles is key to tumor cell autophagy and survival. Through gene expression analyses and cell type abundance estimation in GBMs, we showed that genes associated with the endolysosomal machinery are more prominently featured in non-tumor cells in GBMs than in tumor cells, and that tumor-associated macrophages represent the primary immune cell type that contributes to this trend. Further analyses found an enrichment of endolysosomal pathway genes in immunosuppressive (pro-tumorigenic) macrophages, such as M2-like macrophages or those associated with worse prognosis in glioma patients, but not in those linked to inflammation (anti-tumorigenic). Specifically, genes critical to the hydrolysis function of endolysosomes, including progranulin and cathepsins, were among the most positively correlated with immunosuppressive macrophages, and elevated expression of these genes is associated with worse patient survival in GBMs. Together, these results implicate the hydrolysis function of endolysosomes in shaping the immunosuppressive microenvironment of GBM. We propose that targeting endolysosomes, in addition to its detrimental effects on tumor cells, can be leveraged for modulating immunosuppression to render GBMs more amenable to immunotherapies.

## Introduction

Glioblastoma (GBM), the most common primary malignant brain tumor that accounts for 54% of all glioma cases in the United States [1], has long been recognized as one of the most aggressive and life-threatening cancers. Despite surgical resection, chemotherapy, radiation therapy, and most recently, administration of tumor-treating fields (TTFs), the prognosis of GBM patients remains disappointing, with a median overall survival of 20 months [2]. The

**Funding:** This work was supported by the Preston Robert Tisch Brain Tumor Center and the Department of Pathology at Duke University, the National Institute of Neurological Disorders and Stroke (NINDS) at the National Institutes of Health (NIH) (NS101074), and a pilot research grant from Duke Cancer Institute as part of the NIH National Cancer Institute P30 Cancer Center Support Grant (Grant ID: NIH CA014236).

**Competing interests:** The authors have declared that no competing interests exist.

immunosuppressive tumor microenvironment represents one of the hallmarks of GBM and poses formidable challenges in therapeutic development, manifested by their unresponsiveness to multiple types of immunotherapies [3]. Widely regarded as an immunologically cold tumor, GBMs display an overrepresentation of immunosuppressive cells, such as regulatory T-cells (Tregs), myeloid-derived suppressor cells (MDSCs), and tumor-associated macrophages and microglia (TAMs) [3]. Specifically, TAMs represent the most abundant immune cells that constitute the tumor microenvironment in GBM, and have been recognized to play pivotal roles in shaping the immunosuppressive milieu and promoting tumor progression. The development of TAM-targeting immunotherapies, however, was severely hindered by the phenotypic heterogeneity of TAMs in GBM. These heterogeneous TAM subpopulations have been classified into the pro-inflammatory/anti-tumorigenic M1-like TAMs and the immunosuppressive/pro-tumorigenic M2-like TAMs. Alternatively, the TAM subpopulations responsible for driving immunotherapy resistance have also been categorized using GBM immunosuppressive biomarkers (e.g., *SIGLEC9*, *MARCO*, *SEPP1*). It is therefore imperative to elucidate the mechanisms underlying immunosuppressive TAM polarization and maintenance to devise effective strategies targeting the immunosuppressive TAMs and reinvigorating anti-tumor immunity.

The endolysosomal system has been shown to be essential for cancer cell proliferation, survival, and autophagy [4, 5], and thus has been extensively investigated as a therapeutic target for cancer treatments. Lysosomes are hydrolase-rich intracellular organelles critical for normal development and cell physiology in mammalian cells [6]. When fused with intracellular membrane-enclosed organelles, such as autophagosomes and endosomes, the fused compartments (i.e., autophagolysosomes and endolysosomes) provide the acidic environment in which hydrolases (e.g., cathepsin proteases [7]) become enzymatically active and break down lipids and proteins to recycle these macromolecules and maintain their homeostasis. Various strategies for disrupting endolysosomal functions have been developed, including inducing lysosome membrane permeabilization (LMP; caused by cationic amphiphilic drugs), targeting autophagy (e.g., chloroquine and its derivative hydroxychloroquine), and inhibiting cathepsins and vacuolar $H^+$-ATPases, which are acidic pH-maintaining lysosome membrane proteins [4, 5]. Such strategies were developed primarily based on the consequences of cancer cell killing, and the efficacy of the approach is frequently centered on cancer cells, leaving their direct effect on non-tumor cells and the immune microenvironment unclear [4, 5, 8, 9]. In this study, we implicated the endolysosomal pathways as contributing to the immunosuppressive macrophage phenotype observed in GBMs. This subsequently exposes the potential of utilizing lysosome-modulating agents to simultaneously suppress tumor cell growth, mitigate the pro-tumorigenic effects of immunosuppressive TAMs, and reactivate innate and adaptive anti-tumor immunity.

## Materials and methods

### Estimation of cell type abundance

The mRNA-seq data (gene expression in transcripts per million (tpm)) from The Cancer Genome Atlas GBM (TCGA-GBM) dataset was obtained from the GDC data portal (https://portal.gdc.cancer.gov/). We included only the data of the 162 GBM patients that were explicitly diagnosed with GBM, while excluding the data from the 5 GBM patients without clear diagnostic reports (labeled as "not reported" by the GDC portal). Estimation of cell type abundance was run from the web interface TIMER 2.0 (http://timer.cistrome.org/) [10] using multiple methods to cross-validate our results. We utilized two deconvolution methods, EPIC (Estimating the Proportions of Immune and Cancer cells, via http://epic.gfellerlab.org) [11] and quanTIseq [12], to estimate the proportion of "uncharacterized cells", which mainly

represents tumor cells, and generate scores reflecting the absolute fraction of each cell type [11, 12]. To validate our findings that the endolysosomal cellular components were significantly enriched in macrophages, we employed another estimation method, the single-sample gene set enrichment analysis (ssGSEA) method xCELL, based on its advantage of encompassing 64 immune cell types, including 12 myeloid cell types [13]. For the proportion estimation of M1 and M2 macrophages, the deconvolution method CIBERSORT (absolute mode) was used based on its coverage of different types of macrophages (M0, M1, and M2) [14].

## Pathway enrichment analysis

Pathway enrichment analysis was performed via ShinyGO (version 0.77: http://ge-lab.org/go/) [15] using the Gene Ontology—Cellular Component database, and validated by independent pathway enrichment analysis performed via g:Profiler [16]. For the pathway enrichment analysis of TCGA GBM samples, all protein-coding genes identified in the mRNA-seq data were used as the background gene set, and a false discovery rate (FDR) cutoff of 0.05 was used to define significantly enriched pathways. For the pathway enrichment analysis of *SIGLEC9*[+] TAMs, differentially expressed genes (FDR < = 0.05) in *SIGLEC9*[+] versus *SIGLEC9*[-] TAMs were obtained from a previous study [17], and the total unique number of genes in the GO-CC database was used as the background gene set for the pathway enrichment analysis.

## Gene correlation and survival analysis

Correlation analysis between the immunosuppressive genes and the endolysosomal machinery genes was performed using the gene expression data (tpm) from the TCGA-GBM mRNA-seq dataset or the CGGA (Chinese Glioma Genome Atlas [18]) gene expression data (tpm) obtained from the glioma dataset visualization web application Gliovis (http://gliovis.bioinfo.cnio.es/) [19]. The association between the gene expression and the survival of patients from the CGGA and TCGA-GBM datasets was obtained from Gliovis [19]. Heatmaps were generated using Heatmapper (http://www.heatmapper.ca; clustering method: single linkage; distance measurement method: Euclidean) [20]. For all statistical analyses for gene expression correlations, p-values were determined using GraphPad, and p < 0.05 was considered statistically significant.

## Results

We sought to pinpoint the cellular components and processes that are most perturbed in glioblastoma (GBM), one of the most lethal cancers, to facilitate the identification of therapeutic targets. We utilized the mRNA-seq data of 162 GBM patients from the TCGA-GBM dataset, and estimated the fractions of tumor cells and various types of immune cells using the EPIC (Estimating the Proportions of Immune and Cancer cells) method [11]. We then correlated cancer cells and the remaining cells (i.e., immune cells, endothelial cells, and cancer-associated fibroblasts (CAFs)) with the expression level of each gene (n = 19961 protein-coding genes were examined). Genes that were most positively correlated with cancer cells (cutoff used: r > = 0.3, p < = 0.0001; n = 1257 genes) were subjected to pathway enrichment analyses using the Gene Ontology-Cellular Component (GO-CC) database. The results indicated that, as expected, the most prominent cellular components in cancer cells included those involved in DNA replication (e.g., replisome), gene transcription (e.g., RNA polymerase core complex), protein translation (e.g., tRNA synthetase complex and small ribosomal subunits), and epigenetic reprogramming (e.g., histone acetyltransferase complex and methylosome), all of which were cellular machinery essential to the malignant properties of tumor cells (**Fig 1A**). Interestingly, similar analyses of genes most negatively correlated with cancer cells (r < = -0.5,

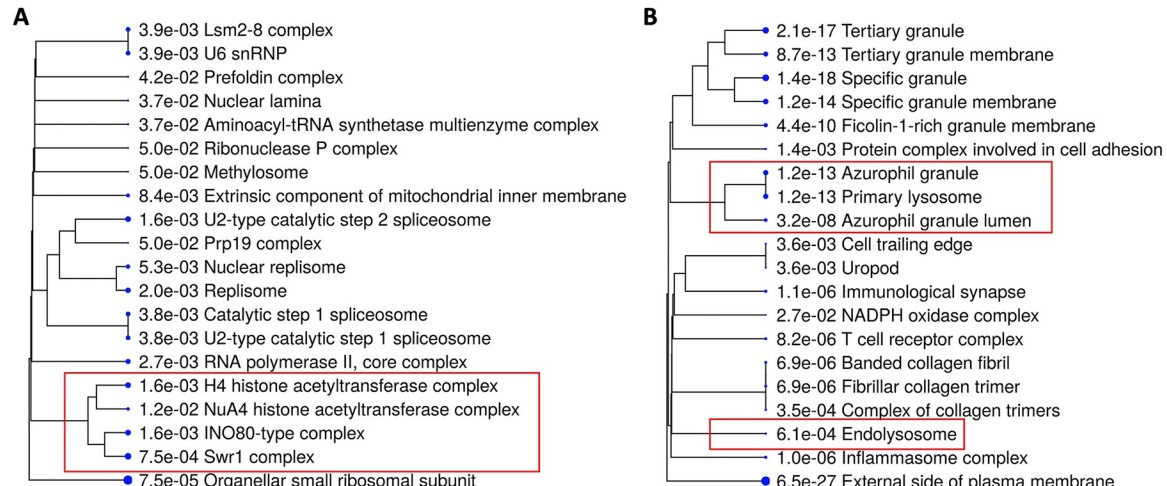

**Fig 1. Gene expression analysis suggests tumor and non-tumor cells in GBMs featured distinct cellular components.** (A-B) The mRNA-seq data (protein-coding genes, tpm) of 162 TCGA GBM samples were subjected to EPIC analysis to estimate the fraction of each cell type. To identify genes that positively or negatively correlated with cancer cells, we calculated each gene's correlation coefficient (r) between the gene expression level and the percentage of cancer cells in each GBM sample. The correlation coefficient cutoff for genes positively or negatively correlated with cancer cells was r > = 0.3 (n = 1257) and r < = -0.5 (n = 654), respectively. (A) GO-Cellular Component enrichment analysis of genes positively correlated with cancer cells (annotated as "Other Cells"). (B) GO-Cellular Component enrichment analysis of genes negatively correlated with cancer cells. (A-B) Dendrograms show the top 20 enriched cellular components sorted by fold enrichment, and the size of each dot denotes the relative number of genes in each cellular component.

p < 0.00001; n = 654) identified an entirely different list of cellular components, including those associated with tumor microenvironment (e.g., protein complex of cell adhesion and collagen trimers), granules and granule membranes, and most notably, primary lysosomes and endolysosomes (**Fig 1B**). These results were corroborated by a subsequent analysis using another deconvolution method, quanTIseq [12] (**S1A and S1B Fig**), suggesting that the endolysosome system not only represents a promising therapeutic target against cancer cells, but might also play significant roles in orchestrating non-tumor cells in GBMs.

Among non-tumor cells identified in the deconvolution analysis, three types of cells were among the most abundant: macrophages, endothelial cells, and CAFs (**S2A Fig**). To assess which of these three cell types contributed to the above enrichment of lysosomes/endolysosomes, we further performed GO-Cellular Component enrichment analysis for genes positively correlated with each cell type. We found that genes most positively correlated with macrophages (r > = 0.5, p < 0.00001; n = 543) were enriched for lysosomes/endolysosomes (**Fig 2A**), and these cellular components were similarly identified when only the genes with the strongest positive correlation (r > = 0.7, p < 0.00001; n = 208) were included for pathway enrichment analysis (**Fig 2B**). In contrast, similar analyses applied to endothelial cells and CAFs did not identify endolysosomal compartments, while cellular components characteristic of each cell type, such as apical plasma membrane and collagen-containing extracellular matrix, were enriched as expected (**S2B and S2C Fig**). Finally, the enrichment of lysosomes/endolysosomes in macrophages was further validated by independent estimation of cell type abundance using another deconvolution method (xCELL) [13] and subsequent pathway analysis (**S2D Fig**). Collectively, these results suggest that lysosomes/endolysosomes are prominent cellular components of GBM tumor-associated macrophages (TAMs), the predominant tumor-infiltrating immune cells that are key to shaping the tumor microenvironment and mediating immunosuppression in GBM [21–23].

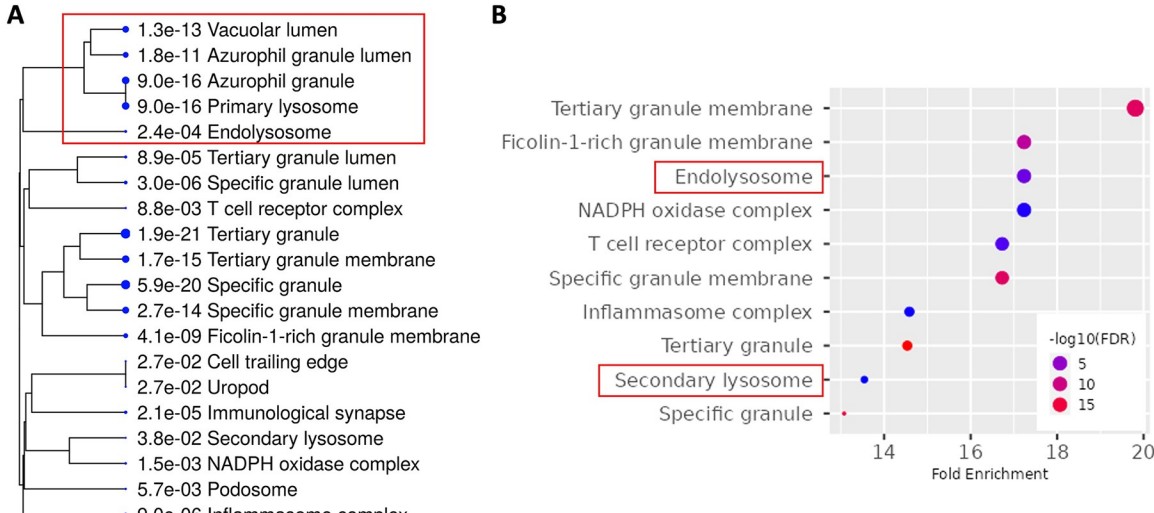

**Fig 2. Tumor-associated macrophages are the major cell type contributing to the endolysosomal feature of GBMs.** (A-B) The mRNA-seq data (protein-coding genes, tpm) of 162 GBM samples were subjected to EPIC analysis to estimate the fraction of each cell type. To identify genes that positively or negatively correlated with macrophages, we calculated each gene's correlation coefficient (r) between the gene expression level and the percentage of macrophages in each GBM sample. The correlation coefficient cutoff for genes positively correlated with macrophages was r > = 0.5 (n = 543), and the cutoff for genes displaying strong positive correlation with macrophages was r > = 0.7 (n = 208). (A) GO-Cellular Component enrichment analysis of genes positively correlated with macrophages. The dendrogram shows the top 20 enriched cellular components sorted by fold enrichment, and the size of each dot denotes the relative number of genes in each cellular component. (B) GO-Cellular Component enrichment analysis of genes identified as displaying strong positive correlations with macrophages. The chart plot shows the top 10 enriched cellular components (FDR cutoff = 0.05, sorted by fold enrichment). The size of each dot corresponds to the fold enrichment for each cellular component as shown on the x-axis.

To provide insight into the regulatory mechanisms of various TAM subtypes in GBM, we further investigated the pathological roles of lysosomes/endolysosomes in different subpopulations of TAMs. For example, if the endolysosomal system is more uniquely associated with the anti-tumorigenic TAMs ("good" TAMs), undesired outcomes may be inflicted when therapeutically disrupting the function of these organelles. In contrast, if they are characteristic of and essential to pro-tumorigenic (i.e., immunosuppressive) TAMs, then targeting lysosomes/endolysosomes can likely mitigate these "bad" cells, providing a beneficial effect in addition to direct killing of tumor cells. To address this question, we first used CIBERSORT to estimate the abundance of two major types of macrophages, M1-like and M2-like macrophages, representing the more anti-tumorigenic and pro-tumorigenic TAMs, respectively [22]. We performed similar cellular component enrichment analysis for genes positively correlated with the fraction of each type of TAM. This analysis revealed that genes positively correlated with M2 macrophages were enriched with components of lysosomes/endolysosomes (S3A Fig). Intriguingly, the cellular components enriched in genes positively correlated with M1 macrophages were dominated by those of cell-cell interactions and cell-extracellular communication (e.g., cell surface proteins and protein complexes involved in cell adhesion) (S3B Fig), suggesting that various types of macrophages likely rely on distinct cellular organelles and compartments to exert their distinct roles in GBMs, such as immunosuppression versus anti-tumor response.

While subtyping TAMs using the M1/M2 paradigm is a useful means of illustrating their opposite roles in cancers, it has emerged that the *in vivo* evidence supporting this dichotomous classification of TAMs remains absent in GBM, and that these cells are more heterogenous than previously thought [21]. Most notably, a recent single-cell RNA sequencing study has

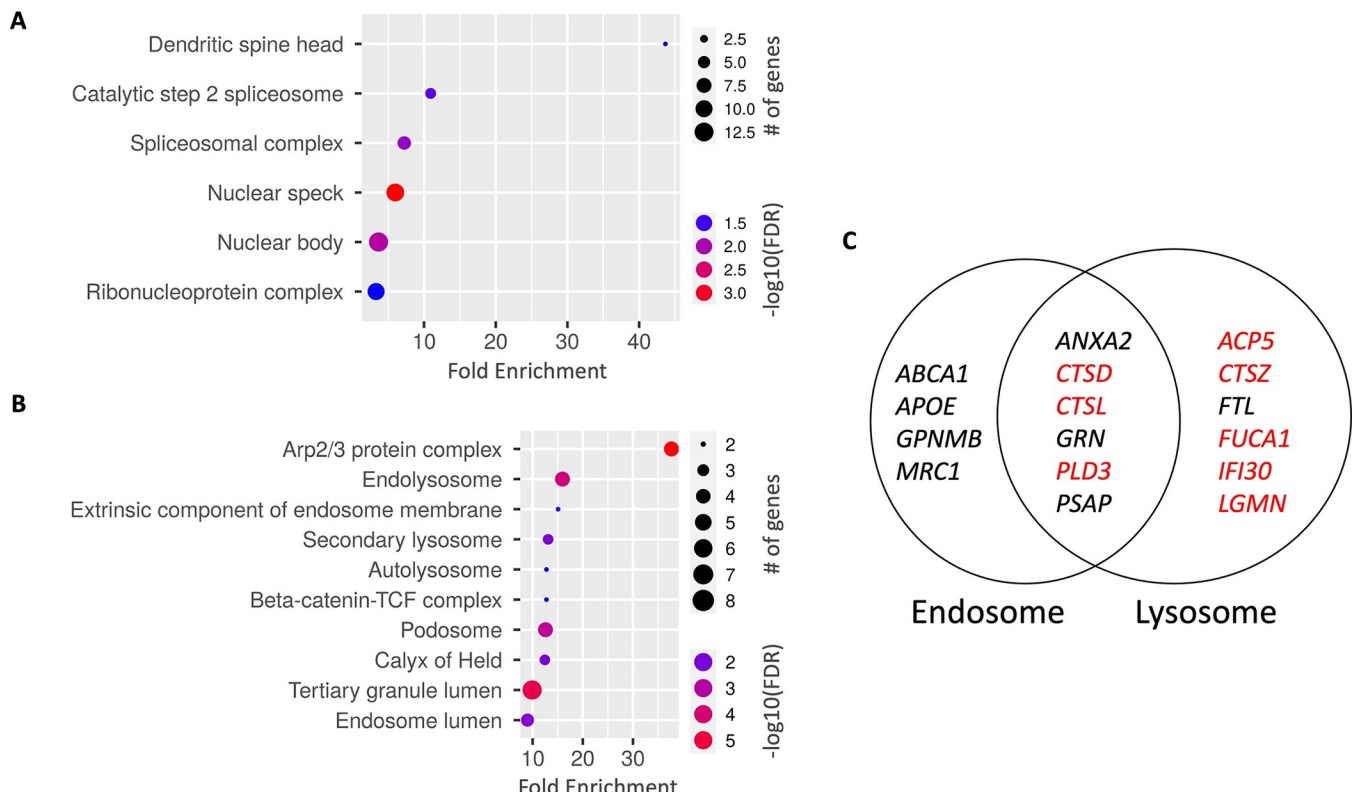

**Fig 3. Myeloid marker genes associated with poor GBM patient survival were enriched for the lysosomes/endolysosomes compartments.** (A) GO-Cellular Component enrichment analysis of 107 pooled marker genes from MC2 and MC7, the two myeloid cell clusters identified to be associated with better prognosis in glioma patients. Six Cellular Components were identified as enriched (FDR cutoff = 0.05, sorted by fold enrichment). (B) GO-Cellular Component enrichment analysis of 309 pooled marker genes from myeloid MC3 and MC5 clusters, the two macrophage clusters identified to be associated with worse prognosis in glioma patients. The chart plot shows the top ten enriched cellular components (FDR cutoff = 0.05, sorted by fold enrichment). (A-B) The size of each dot corresponds to the fold enrichment for each cellular component as shown on the x-axis. (C) Illustration of the 16 MC3 marker genes that encode proteins determined to be localized at endosomes and/or lysosomes with high confidence, as denoted by the COMPARTMENTS database (https://compartments.jensenlab.org/Search). Enzyme-coding genes were labeled in red.

defined nine myeloid clusters (MC) in human gliomas (including low-grade glioma, newly diagnosed GBMs, and recurrent GBMs), and identified those that were correlated with better survival (e.g., MC2 and MC7, microglia clusters proposed to be anti-tumorigenic) or worse survival (e.g., MC3 and MC5, macrophage clusters proposed to be immunosuppressive and pro-tumorigenic) in GBM patients [24]. We performed a similar analysis for marker genes for these two groups of myeloid cells, and found that lysosomes/endolysosomes were enriched only in the pro-tumorigenic myeloid clusters (**Fig 3A and 3B**). Further individual examination of marker genes for MC3 (n = 55) and MC5 (n = 254) found that the enrichment of the lysosome/endolysosome components, while observed in both MCs, was particularly dominant in MC3 (**S3C Fig**), the cluster that was more strongly associated with poor survival in GBM patients [24]. In all, 21 (38%) of the 55 marker genes for MC3 were defined as being explicitly associated with GO-Cellular Components of lysosomes and/or endosomes (GO:0005764, GO:0005768, and GO:0043202), and 16 (29%) of them encode proteins that have been determined to reside in endosomes and/or lysosomes per the COMPARTMENTS database [25], including proteins directly involved in the degradation process in endolysosomes, such as hydrolases (e.g., *CTSD*, *CTSL*, *CTSZ*, and *FUCA1*) and progranulin (*GRN*), a non-enzymatic protein recently found to be indispensable for lysosomal hydrolysis [26, 27] (**Fig 3C** and

S1 Table). An in-depth examination of the MC3 genes further corroborates the roles of endolysosomes in GBM pathogenesis and, specifically, in immunosuppression. First, we assessed the relationship between the 21 MC3 genes and a set of four genes representative of GBM's immunosuppressive microenvironment: the immunosuppressive TAM markers *MRC1* (CD206) and *CD163* [21] and the immunosuppressive cytokines *IL10* and *TGFB1* [21, 23], and found overwhelmingly positive correlations between these two groups of genes (**Fig 4A and 4B**, **S4 Fig,** and **S1 Table**). Subsequently, we examined the association of each of the 21 MC3 genes with GBM patient survival from the TCGA and CGGA datasets, and found that the high expression level of each of the 11 genes was significantly associated with worse overall survival in at least one of the datasets (**S1 Table**). Among them, six genes were found to be significantly linked to worse survival in the TCGA dataset while displaying similarly significant association or at least a consistent trend in the CGGA dataset, including five genes that encode proteins that primarily localized in endolysosomes (*CTSD*, *CTSZ*, *FUCA1*, *GRN*, and *IFI30*) (**Fig 4C and 4D** and **S1 Table**).

While the findings above suggest a role of endolysosomes in GBM pathogenesis, particularly in shaping GBM's immunosuppressive microenvironment, they raise the question of the roles of endolysosomes in the context of GBM resistance to immunotherapy. To address this question, we took advantage of the recent findings that subsets of *SIGLEC9*+ TAMs dampen GBM's response to immunotherapy [17, 28]. We specifically focused on two subsets of *SIGLEC9*+ TAMs that were found to be highly plastic and immunosuppressive (*SIGLEC9*+*MARCO*+ TAMs, cluster C9, and *SIGLEC9*+*SEPP1*+ TAMs, cluster C2) [17], and examined genes that were upregulated in these two populations of TAMs (versus their respective *SIGLEC9*- counterparts). Enrichment analysis of GO-Cellular Components revealed that in each case, lysosomes/endosomes were enriched only in genes that were upregulated in *SIGLEC9*+ TAMs, but not those in their *SIGLEC9*- counterparts (**S5A and S5B Fig**). Independently, KEGG pathway analysis of genes that defined the *SIGLEC9*+ TAMs confirmed lysosomes as the most prominent feature of these immunosuppressive TAMs (**S5C Fig**). Of note, the five genes described above (*CTSD*, *CTSZ*, *FUCA1*, *GRN*, and *IFI30*) were among those significantly elevated in the immunosuppressive *SIGLEC9*+ TAMs [17]. Together with previous findings [17, 28], the results from these analyses suggest that endolysosomes likely play instrumental roles in conferring GBM resistance to immunotherapy.

## Discussion

### Past and present: Endolysosomes and immunosuppressive TAMs

In summary, by analyzing gene expression profiles and performing cell type deconvolution and cellular component enrichment analyses in subsets of non-tumor cells in GBMs, we postulate that the lysosomal machinery is a major component associated with immunosuppressive (pro-tumorigenic) TAMs. In addition to their well-documented roles in tumor cells, lysosomes also promote GBM pathogenesis and resistance to immunotherapy through a non-tumor cell-autonomous manner. While these findings were primarily based on suggestive results from gene expression and correlative analyses, they corroborate prior independent studies. For instance, chloroquine, through disrupting the function of lysosomes, stimulates anti-tumor immunity in a melanoma tumor model by switching M2 macrophages toward the M1 phenotype [29]. Although the microenvironments in GBM and melanoma are expected to be substantially different, the chloroquine-driven M2 to M1 switch provides direct evidence to support the notion that functionally opposite types of macrophages indeed rely differently on the endolysosomal function. Additionally, it is noted that neurodegenerative conditions in humans caused by loss-of-function mutations of genes in endolysosomal machinery

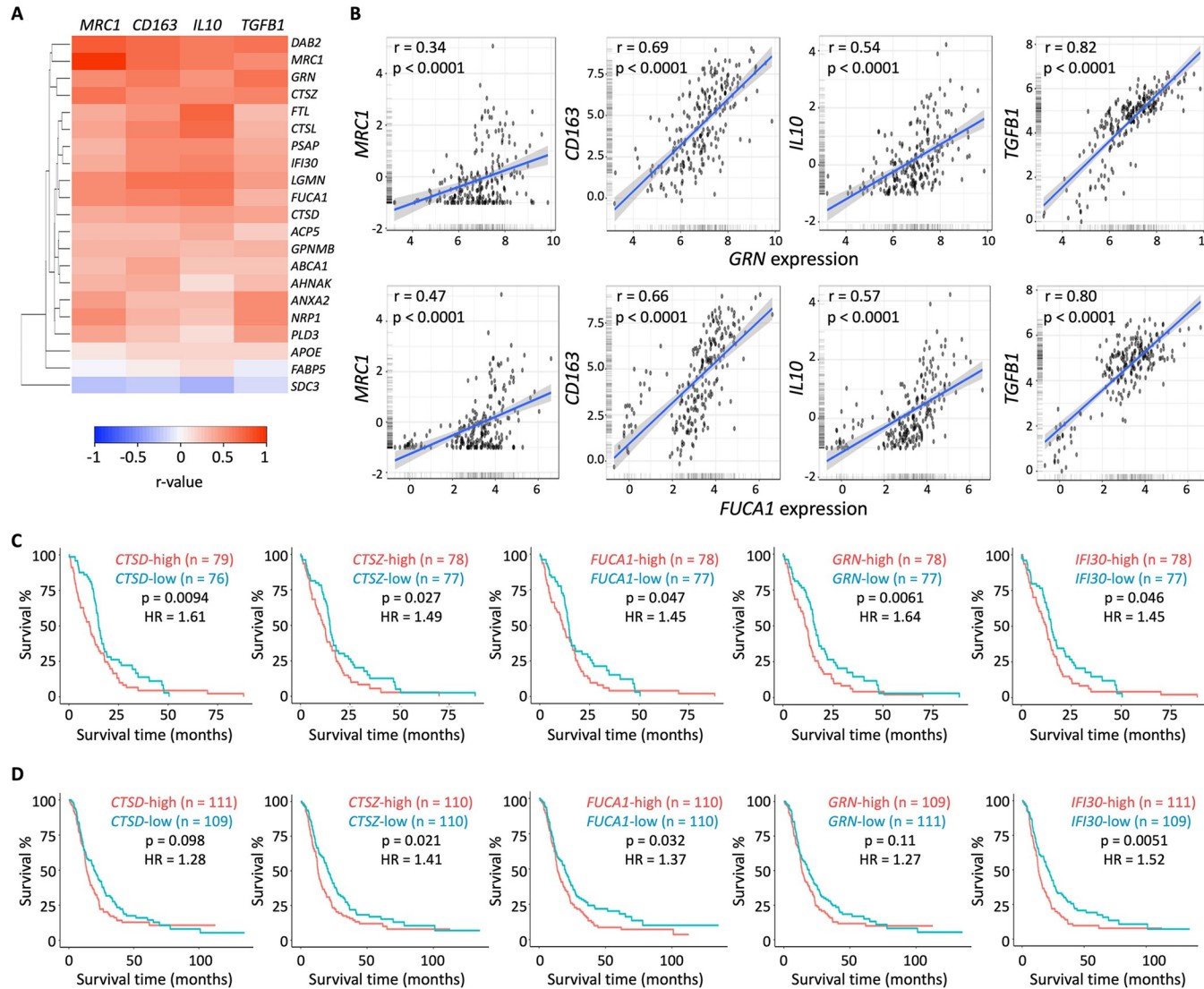

**Fig 4. Endolysosomal genes correlate with immunosuppressive marker genes and are associated with worse GBM patient survival.** (A) Heatmap demonstrating the Pearson correlation coefficient values (r-values) between the expression of 21 endolysosomal genes and the immunosuppression markers *MRC1* (CD206), *CD163*, *IL10*, and *TGFB1*. The heatmap was generated using Heatmapper (clustering method: single linkage; distance measurement method: Euclidean). The TCGA GBM mRNA-seq dataset was used for the correlation analyses (n = 162). (B) Scatter plots showing examples of positive correlations between the expression of endolysosomal genes and immunosuppression markers. The CGGA primary GBM dataset was used for the analyses (n = 223). Plots were generated using the GlioVis portal, and r-values and p-values were calculated based on Pearson correlation analysis by the Gliovis portal. (C) Kaplan-Meier survival analyses indicating significant correlations between the expression of five endolysosomal genes and GBM patients' overall survival. The TCGA GBM mRNA-seq dataset was used for the analyses (n = 155). (D) Kaplan-Meier survival analyses indicating significant correlations between the expression of five endolysosomal genes and GBM patients' overall survival. The CGGA primary GBM dataset was used for the analyses (n = 220). (C-D) Median expression was used as the cutoff value for high versus low gene expression. Plots were generated using the GlioVis portal, and p-values and hazard ratios were calculated based on log-rank tests and the Cox proportional hazards model, respectively, by the Gliovis portal.

commonly feature microglia activation (i.e., neuroinflammation) [30, 31]. The most intriguing examples include *GRN* and *FUCA1*, as their loss of functions directly causes lysosomal dysregulation and neuroinflammation, accompanied by neurodegenerative disorders such as Alzheimer's disease (AD) and frontotemporal dementia (FTD) [32–34]. Separately, *GRN* has been known to be an immunosuppressive protein [35], and a recent study has linked *FUCA1* to autophagy and macrophage infiltration in gliomagenesis [36].

The mechanistic link between the endolysosomal machinery and immunosuppressive TAM polarization/maintenance represents another gap in knowledge in the development of GBM immunotherapy. Previous literature on physiological macrophage polarization has demonstrated the functional importance of the endolysosomal compartments in alternative activation (M2) of macrophages. For example, lysosomal lipolysis by LPL, a hydrolytic enzyme requiring the acidic pH in lysosomes, has been shown to play an essential role in the activation and survival of M2 macrophages [37], and the lysosomal amino acid-sensing complex, which consists of Lamtor1 and v-ATPase, was also demonstrated to be indispensable in M2 polarization [38]. Moreover, mitigating lysosomal acidity and lysosomal protease activity can promote a phenotypic switch towards anti-tumorigenic M1-like macrophages in tumor-bearing mouse models [39]. Therefore, we speculate that similar machinery might also contribute to the association between lysosomes and immunosuppressive TAM polarization/maintenance in GBM. These immunosuppressive TAMs can subsequently shape the immunosuppressive tumor microenvironment and promote immunotherapy resistance in GBM through mechanisms such as (i) secreting immunosuppressive cytokines (e.g., IL-6, IL-10, TGF-$\beta$) (ii) promoting T cell exhaustion, and (iii) secreting chemoattractant to recruit regulatory T cells (Tregs) and myeloid-derived suppressor cells (MDSCs) [40, 41], as indicated by previous literature.

## Limitations

Our study has several notable limitations. First, most of the findings included in this study are supported by association-based approaches, and further research needs to be conducted to investigate the functional role of endolysosomes in TAM polarization and maintenance. Second, for our gene expression correlation analyses, it is of critical importance to consider the confounding variables that could affect the accurate interpretation of the correlation results, such as the different GBM subtypes (e.g., proneural, classical, mesenchymal), genders, and the heterogeneous genetic background of GBM tumors. Nevertheless, the association between the endolysosomal machinery and immunosuppressive TAMs presented in this study was independently validated through various deconvolution methods and pathway enrichment analyses, and thus warrants further investigation into the therapeutic potential of endolysosome-targeting strategies in GBM.

## Future implications

The findings presented in this study also inform our understanding of TAMs and have direct implications for targeting the endolysosomal system for cancer treatments in several ways. While the heterogeneity and plasticity of TAMs, as defined by their distinct gene expression profiles / featured molecular pathways (e.g., Hallmark pathways), have been well documented in GBMs, they can also be viewed in a highly simplified way as two functionally opposite types (i.e., immunosuppressive/pro-tumorigenic and immunoactive/anti-tumorigenic) that produce different cytokines and respond to cytokines in different ways [21, 24]. Our cellular component-focused findings suggest that heterogenous and functionally opposite TAMs also depend on distinct cellular organelles for exerting their pro- or anti-tumorigenic effects. This suggests that although lysosomal hydrolysis has been known to be a cellular process critical to macrophages, it is possible that therapeutic targeting of lysosomes can affect immunosuppressive TAMs more than immunoactive TAMs, and shift the TAMs and the GBM microenvironment toward a state favorable for anti-tumor response and immunotherapy. We propose that, while the relationship between endolysosomes and the TAMs in GBMs awaits further investigation, it is warranted to incorporate assays of immune cells and immune microenvironment when

assessing endolysosome-based therapeutic approaches in translational research and clinical trials (e.g., chloroquine [42]) for GBM patients.

Multiple strategies for targeting lysosomes have been devised, including altering the acidic microenvironment (e.g., chloroquine), damaging lysosomal membrane (e.g., cationic amphiphilic drugs), and direct inhibition of hydrolases [9]. In this context, progranulin is particularly interesting. While progranulin is a secreted protein, it can be internalized into cells through cognate receptors and processed into the functional form, termed granulin, in endolysosomes to support the hydrolysis function of these organelles [43–45]. Most importantly, recent studies have revealed that the major mechanistic role of granulin in endolysosomes is to maintain the homeostasis of an unusual, endolysosome-specific family of lipids, bis(monoacylglycero) phosphate (BMP), in these organelles [26, 27]. Thus, multiple proteins in this progranulin-BMP axis can be explored for therapeutically targeting the progranulin pathway. Of note, a previous study also suggests that progranulin promotes GBM cell stemness and their resistance to temozolomide, raising the intriguing possibility that targeting the progranulin pathway can provide dual benefits: attenuation of both GBM stemness and immunosuppression [46].

## Supporting information

**S1 Fig. Gene expression analysis suggests tumor and non-tumor cells in GBMs featured distinct cellular components.** (A-B) The mRNA-seq data (protein-coding genes, tpm) of 162 TCGA GBM samples were subjected to quanTIseq analysis to estimate the fraction of each cell type. To identify genes that positively or negatively correlated with cancer cells, we calculated each gene's correlation coefficient (r) between the gene expression level and the percentage of cancer cells in each GBM sample. The correlation coefficient cutoff for genes positively or negatively correlated with cancer cells was r > = 0.3 (n = 290) and r < = -0.5 (n = 347), respectively. (A) GO-Cellular Component enrichment analysis of genes positively correlated with cancer cells (annotated as "Other Cells"). (B) GO-Cellular Component enrichment analysis of genes negatively correlated with cancer cells (annotated as "Other Cells"). (A-B) Dendrograms show the top 20 enriched cellular components sorted by fold enrichment, and the size of each dot denotes the relative number of genes in each cellular component.
(TIF)

**S2 Fig. Tumor-associated macrophages are the major cell type contributing to the endolysosomal feature of GBMs.** (A-C) The mRNA-seq data (protein-coding genes, tpm) of 162 TCGA GBM samples were subjected to EPIC analysis to estimate the fraction of each cell type. (A) Box plot showing the fractions of different cell types, including infiltrating immune cells and other cells (i.e., cancer cells), as estimated by EPIC. To identify genes that positively correlated with endothelial cells or CAF, we calculated each gene's correlation coefficient (r) between the gene expression level and the percentage of endothelial cells or CAF in each GBM sample. The correlation coefficient cutoff for genes positively correlated with endothelial cells was r > = 0.3 (n = 237), and the cutoff for genes displaying strong positive correlation with CAF was r > = 0.5 (n = 543). (B) GO-Cellular Component enrichment analysis of genes positively correlated with endothelial cells. (C) GO-Cellular Component enrichment analysis of genes positively correlated with CAF. (D) The mRNA-seq data (protein-coding genes, tpm) of 162 TCGA GBM samples were subjected to xCELL analysis to estimate the fraction of each cell type, and genes positively correlated with macrophage (r > = 0.5) were used for GO-Cellular Component enrichment analysis. (B-D) Dendrograms show the top 20 enriched cellular components sorted by fold enrichment, and the size of each dot denotes the relative number of genes in each cellular component.
(TIF)

**S3 Fig. Different subsets of myeloid cells featured distinct cellular components.** (A-B) The mRNA-seq data (protein-coding genes, tpm) of 162 TCGA GBM samples were subjected to CIBERSORT (absolute mode) analysis to estimate the fractions of M1 and M2 macrophages. To identify genes that positively correlated with M1 or M2 macrophages, we calculated each gene's correlation coefficient (r) between the gene expression level and the percentage of M1 or M2 macrophages in each GBM sample. The correlation coefficient cutoff for genes positively correlated with M1 macrophages was $r >= 0.3$ (n = 318), and the cutoff for genes displaying strong positive correlation with M2 macrophages was $r >= 0.5$ (n = 429). Note that the cutoff value of $r >= 0.3$ was used for the M1 macrophage analysis to have the number of analyzed genes more comparable to the M2 macrophage analysis (when the cutoff value of $r >= 0.5$ was used, only 56 genes were identified). (A) GO-Cellular Component enrichment analysis of genes positively correlated with M2 macrophages. (B) GO-Cellular Component enrichment analysis of genes positively correlated with M1 macrophages. (A-B) Dendrograms show the significantly enriched cellular components (FDR cutoff = 0.05), and the size of each dot denotes the relative number of genes in each cellular component. (C) GO-Cellular Component enrichment analysis of marker genes for MC3 or MC5. The table shows the top 20 enriched cellular components sorted by the number of genes identified. Cellular components directly indicative of lysosomes/endosomes were highlighted in red.
(TIF)

**S4 Fig. Endolysosomal machinery genes correlate with marker genes indicative of immunosuppression.** Heatmap demonstrating the Pearson correlation coefficient values (r-values) between the expression of 21 endolysosomal genes and immunosuppression markers MRC1 (CD206), CD163, IL10, and TGFB1. The heatmap was generated using Heatmapper (clustering method: single linkage; distance measurement method: Euclidean). The CGGA primary GBM dataset was used for the correlation analysis (n = 223).
(TIF)

**S5 Fig. Lysosomes/endosomes were featured in *SIGLEC9*[+] TAMs but not in their *SIGLEC9*[-] counterparts.** (A) GO-Cellular Component enrichment analysis of genes upregulated in *SIGLEC9*[+]*MARCO*[+] TAMs (left panel) or *SIGLEC9*[-]*MARCO*[-] TAMs (right panel). (B) GO-Cellular Component enrichment analysis of genes upregulated in *SIGLEC9*[+]*SEPP1*[+] TAMs (left panel) or *SIGLEC9*[-]*SEPP1*[-] TAMs (right panel). (A-B) Dendrograms show the top 20 cellular components sorted by the number of genes identified. Cellular components directly indicative of lysosomes/endosomes were highlighted in red. (C) KEGG pathway analysis of genes upregulated in *SIGLEC9*[+]*SEPP1*[+] TAMs. The chart plot shows the top 10 enriched pathways (FDR cutoff = 0.05, sorted by fold enrichment). Similar analyses of genes upregulated in *SIGLEC9*[+]*MARCO*[+] TAMs also identified lysosomes as significantly enriched. All known protein-coding genes were used as the background gene set. (A-C) For identifying upregulated genes, an FDR cutoff value of 0.05 was used for all analyses.
(TIF)

**S1 Table. Twenty-one MC3 marker genes were associated with the GO-cellular component of endosomes and/or lysosomes.** The endosome/lysosome localization of each protein was defined by the COMPARTMENTS (Jensen) database. For survival analyses, the TCGA GBM mRNA-seq dataset (n = 155) and CGGA primary GBM dataset were used (n = 220) (gene expression low versus high cutoff: median expression; significance: log-rank p-value $<= 0.05$; n.a.: no information available for CTSL in the TCGA dataset). *Genes that have the following characteristics: endosome/lysosome is the most or one of the most likely localizations, and the

correlation between expression and survival was significant or showing consistent trend in both TCGA and CGGA GBM datasets. **The Pearson correlation coefficients (r-values) between the expression (tpm) of designated genes were calculated using the TCGA GBM mRNA-seq dataset (n = 162).

(XLSX)

## Author Contributions

**Conceptualization:** Michael A. Sun, Haipei Yao, Vidyalakshmi Chandramohan, David M. Ashley, Yiping He.

**Data curation:** Michael A. Sun, Qing Yang, Vidyalakshmi Chandramohan, Yiping He.

**Formal analysis:** Michael A. Sun, Qing Yang, Yiping He.

**Funding acquisition:** Yiping He.

**Investigation:** Michael A. Sun, Haipei Yao, Vidyalakshmi Chandramohan, Yiping He.

**Methodology:** Michael A. Sun, Haipei Yao, Qing Yang, Vidyalakshmi Chandramohan, Yiping He.

**Resources:** Yiping He.

**Supervision:** Yiping He.

**Validation:** Michael A. Sun, Yiping He.

**Writing – original draft:** Michael A. Sun, Haipei Yao, Qing Yang, Christopher J. Pirozzi, Vidyalakshmi Chandramohan, David M. Ashley, Yiping He.

**Writing – review & editing:** Michael A. Sun, Haipei Yao, Qing Yang, Christopher J. Pirozzi, Vidyalakshmi Chandramohan, David M. Ashley, Yiping He.

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
