## [Decision Letter · Decision Letter 0]

14 Nov 2023

PONE-D-23-33821Gene expression analysis suggests immunosuppressive roles of endolysosomes in glioblastomaPLOS ONE

Dear Dr. He,

Thank you for submitting your manuscript to PLOS ONE. After careful consideration, we feel that it has merit but does not fully meet PLOS ONE’s publication criteria as it currently stands. Therefore, we invite you to submit a revised version of the manuscript that addresses the points raised during the review process.

We look forward to receiving your revised manuscript.

Kind regards,

Syed M. Faisal, Ph.D.

Academic Editor

PLOS ONE

Journal Requirements:

2. For studies reporting research involving human participants, PLOS ONE requires authors to confirm that this specific study was reviewed and approved by an institutional review board (ethics committee) before the study began. Please provide the specific name of the ethics committee/IRB that approved your study, or explain why you did not seek approval in this case.

 "This work was supported by the Preston Robert Tisch Brain Tumor Center and the Department of Pathology at Duke University, the National Institute of Neurological Disorders and Stroke (NINDS) at the National Institutes of Health (NIH) (NS101074), and a pilot research grant from Duke Cancer Institute as part of the NIH National Cancer Institute P30 Cancer Center Support Grant (Grant ID: NIH CA014236)."

Reviewers' comments:

Reviewer's Responses to Questions

**Comments to the Author**

1. Is the manuscript technically sound, and do the data support the conclusions?

Reviewer #1: Yes

Reviewer #2: Yes

2. Has the statistical analysis been performed appropriately and rigorously? 

Reviewer #1: Yes

Reviewer #2: Yes

3. Have the authors made all data underlying the findings in their manuscript fully available?

Reviewer #1: Yes

Reviewer #2: Yes

4. Is the manuscript presented in an intelligible fashion and written in standard English?

Reviewer #1: Yes

Reviewer #2: Yes

5. Review Comments to the Author

Reviewer #1: The manuscript entitled “Gene expression analysis suggests immunosuppressive roles of endolysosomes in glioblastoma” shows that genes associated with the endolysosomal machinery are more prominently featured in non-tumor cells in GBMs than in tumor cells, and that tumor-associated macrophages represent the primary immune cell type that contributes to this trend. Further the authors found an enrichment of endolysosomal pathway genes in immunosuppressive (pro-tumorigenic) macrophages, such as M2-like macrophages or those associated with worse prognosis in glioma patients, but not in those linked to inflammation (anti-tumorigenic). Specifically, genes critical to the hydrolysis function of endolysosomes, including progranulin and cathepsins, were among the most positively correlated with immunosuppressive macrophages, and elevated expression of these genes is associated with worse patient survival in GBMs. Based on results the authors conclude that the hydrolysis function of endolysosomes is important in shaping the immunosuppressive microenvironment of GBM. The authors propose that targeting endolysosomes, in addition to its detrimental effects on tumor cells, can be leveraged for modulating immunosuppression to render GBMs more amendable to immunotherapies. The manuscript needs minor revision before it can be accepted for publication.

1. The introduction needs to be changed. It is too small and less informative.

2. The english grammer needs to be worked upon.

Reviewer #2: The study examines how lysosomes are involved in glioblastoma and their role in immunotherapy resistance. They suggest that lysosomes are linked to immunosuppressive tumor-associated macrophages (TAMs), which may contribute to glioblastoma's resistance to immunotherapy. The article offers valuable insights but needs some structural improvements.

Methodology and Data Analysis:

Could you provide more details on the selection criteria for the 162 GBM patients from the TCGA-GBM dataset? Were any specific criteria applied for patient inclusion or exclusion?

In the methodology section, you mention the use of multiple deconvolution methods. Could you explain why multiple methods were employed, and what advantages each method offers in this context?

The pathway enrichment analysis was conducted using ShinyGO and validated by g:Profiler. Could you elaborate on the specific parameters and settings used for these analyses?

Were any steps taken to account for potential confounding factors when analyzing gene expression correlations in the glioma dataset visualization?

Results and Interpretation:

The study suggests that lysosomes are associated with GBM resistance to immunotherapy. Could you provide more insights into the mechanisms by which lysosomes contribute to immunotherapy resistance?

Could you discuss the potential implications of the findings for clinical practice and the development of new therapeutic approaches for GBM?

Discussion and Limitations:

10. The discussion section could benefit from an explicit discussion of the study's limitations. What limitations or potential sources of bias should readers be aware of when interpreting the results?

How do the findings of this study align with or differ from previous research on lysosomes and their role in cancer, particularly in GBM?

Organization and Presentation:

Would you consider adding more structured subheadings to the discussion section to enhance the organization of your arguments and conclusions?

Could you provide more context for the figures and tables in the article? Some readers may benefit from additional explanations in the figure captions.

6. PLOS authors have the option to publish the peer review history of their article (what does this mean?). If published, this will include your full peer review and any attached files.

Reviewer #1: **Yes: **Sidra Islam

Reviewer #2: No

---

## [Author Response · Author response to Decision Letter 0]

23 Jan 2024

Response to Reviewers

Manuscript: Gene expression analysis suggests immunosuppressive roles of endolysosomes in glioblastoma (PONE-D-23-33821)

Journal: PLOS ONE

• Overall comments:

Reviewer 1:

The manuscript needs minor revision before it can be accepted for publication.

1. The introduction needs to be changed. It is too small and less informative.

2. The english grammar needs to be worked upon.

Reviewer 2:

The article offers valuable insights but needs some structural improvements.

Response: We thank the reviewers for their positive comments and constructive suggestions for improving the manuscript. We have addressed the comments by expanding the introduction to include additional information and have also double-checked the language. We have revised the manuscript to improve its overall structure. More specific revisions are described in the following sections. 

• Methodology and Data Analysis:

1. Could you provide more details on the selection criteria for the 162 GBM patients from the TCGA-GBM dataset? Were any specific criteria applied for patient inclusion or exclusion?

Response: We thank the reviewers for providing us with this important subject of discussion. In this study, we obtained the TCGA-GBM mRNA-seq data using the GDC portal (https://portal.gdc.cancer.gov/) and included only the data of the 162 GBM patients that were explicitly diagnosed with GBM, while excluding the data from the 5 GBM patients without clear diagnostic reports (labeled as “not reported” by the GDC portal). We have inserted this information in the Methods section of the manuscript.

2. In the methodology section, you mention the use of multiple deconvolution methods. Could you explain why multiple methods were employed, and what advantages each method offers in this context?

Response: We appreciate the reviewers for highlighting this important question regarding the methodology for the estimation of cell population abundance. The methods employed in this study, including the deconvolution methods EPIC, quanTIseq, and CIBERSORT, along with the single-sample gene set enrichment analysis (ssGSEA) method xCELL, are all well-established methods frequently utilized in peer-reviewed literature for estimating the proportion of tumor-infiltrating immune cells. Among all, EPIC and quanTIseq demonstrate robust overall performance, and are the only two methods that estimate the proportion of “uncharacterized cells”, which mainly represent tumor cells, and generate scores reflecting the absolute fraction of each cell type (1, 2). Therefore, we utilized these two methods to inquire about the cellular pathways enriched in tumor or immune cells (Fig 1A-B and S1A-B Fig). 

To validate our findings that the endolysosomal cellular components were significantly enriched in macrophages, as deconvoluted by the EPIC method (Fig 2A-B and S2A-C Fig), we performed another cell-type abundance estimation using the ssGSEA method xCELL based on its advantage of encompassing 64 immune and stromal cell types (S2D Fig). Specifically, the xCELL method includes gene signatures of 12 myeloid cell types, providing higher predicting power in dissecting the heterogeneous myeloid compartments, including macrophages. Finally, we incorporated the CIBERSORT method into the study due to the reason that our lab has more extensive experience in using the CIBERSORT method for immune cell fraction estimation, and owned pre-established protocols and control datasets for CIBERSORT analysis from our previous publication (3). In addition, CIBERSORT provides the coverage of different types of macrophages (M0, M1, and M2), which are not available in the EPIC reference profiles, making it a superior option for our purpose of identifying pathways specifically enriched in immunosuppressive M2-like macrophages. We briefly inserted the rationale of using various deconvolution methods in the Methods section.

3. The pathway enrichment analysis was conducted using ShinyGO and validated by g:Profiler. Could you elaborate on the specific parameters and settings used for these analyses?

Response: We appreciate the thoughtful question about the experimental details of the pathway enrichment analyses. ShinyGO pathway enrichment analysis was performed using the Gene Ontology - Cellular Component (GO-CC) database with redundant pathways removed by the software. Pathways were considered significantly enriched with a false discovery rate (FDR) cutoff of 0.05, and then sorted by fold enrichment. For the analysis of TCGA GBM samples, all protein-coding genes identified in the mRNA-seq data were used as the background gene set. For the pathway enrichment analysis of SIGLEC9+ TAMs, the total unique number of genes in the GO-CC database was used as the background gene set. Accordingly, we have included the above description in the method section of the revised manuscript.

The analysis results were validated by independent pathway enrichment analysis using the g:GOSt tool of the g:profiler web server. The analysis was similarly performed using the Gene Ontology - Cellular Component (GO-CC) database, and pathways were considered significantly enriched with a g:SCS threshold of 0.05. For the analysis of TCGA GBM samples, all protein-coding genes identified in the mRNA-seq data were used as the background gene set. For the pathway enrichment analysis of SIGLEC9+ TAMs, the total unique number of genes in the GO-CC database was used as the background gene set.

4. Were any steps taken to account for potential confounding factors when analyzing gene expression correlations in the glioma dataset visualization?

Response: We thank the reviewer for raising this discussion point that needs further clarification and allowing us to revisit the experimental limitations of the analysis. The gene expression correlation analysis was initially performed in an unbiased way to identify any potential transcriptional correlations between endolysosomal genes and GBM immunosuppressive markers. Nevertheless, it is of critical importance to consider the potential confounding factors, including the different GBM subtypes (e.g., proneural, classical, mesenchymal), genders, and the heterogeneous genetic background of GBM tumors, to accurately interpret the correlation analysis results. The interpretation of this study’s gene expression correlation analysis is thus limited by the potential confounding effects of the factors mentioned above. Accordingly, we have included a discussion clarifying this point in the revised manuscript by inserting a “Limitations” paragraph in the Discussion section.

• Results and Interpretation:

1. The study suggests that lysosomes are associated with GBM resistance to immunotherapy. Could you provide more insights into the mechanisms by which lysosomes contribute to immunotherapy resistance?

Response: We thank the reviewer for providing us with this crucial line of discussion. In this manuscript, we demonstrated that the endolysosomal components were enriched within the immunosuppressive and immunotherapy-resistant tumor-associated macrophage (TAM) gene signatures in GBM. Mechanistically, we hypothesize that the endolysosomal pathways play pivotal roles in promoting TAM polarization towards immunosuppressive phenotype and/or maintaining the function and survival of M2-like TAMs. These immunosuppressive TAMs can subsequently shape the immunosuppressive tumor microenvironment and promote immunotherapy resistance in GBM through mechanisms such as (i) secreting immunosuppressive cytokines (e.g., IL-6, IL-10, TGF-𝛃) (ii) promoting T cell exhaustion, and (iii) secreting chemoattractant to recruit regulatory T cells (Tregs) and myeloid-derived suppressor cells (MDSCs) (4, 5). 

The mechanistic link between the endolysosomal machinery and immunosuppressive TAM polarization/maintenance is another gap in knowledge in the development of GBM immunotherapy. Previous literature on physiological macrophage polarization has demonstrated the functional importance of the endolysosomal compartments in alternative activation (M2) of macrophages. For example, lysosomal lipolysis by LPL, a hydrolytic enzyme requiring the acidic pH in lysosomes, has been shown to play an essential role in the activation and survival of M2 macrophages (6), and the lysosomal amino acid-sensing complex, which consists of Lamtor1 and v-ATPase, was also demonstrated to be indispensable in M2 polarization (7). Moreover, mitigating lysosomal acidity and lysosomal protease activity can directly promote a phenotypic switch towards anti-tumorigenic M1-like macrophages in tumor-bearing mouse models (8). Therefore, we speculate that similar machinery might also contribute to the association between lysosomes and immunotherapy resistance in GBM. We have inserted the discussion of tentative mechanisms briefly in the Discussion section of the manuscript.

2. Could you discuss the potential implications of the findings for clinical practice and the development of new therapeutic approaches for GBM?

Response: We thank the reviewer for providing these key discussion points that enable us to broaden the scope of this manuscript and present our findings in a bigger picture. Despite the fact that TAMs have been identified to be the major immune cell type responsible for the immunosuppressive microenvironment and immunotherapy resistance in GBM, the heterogeneous subpopulations of TAMs remain the primary obstacle to the development of effective TAM-targeting immunotherapy. Our findings implicated the endolysosomal pathways in the immunosuppressive macrophage phenotype, opening the possibility of utilizing lysosome-modulating agents to simultaneously suppress tumor cell growth, mitigate the pro-tumorigenic effects of immunosuppressive TAMs, and reactivate innate and adaptive anti-tumor immunity. We have inserted a subsection titled “Future implications” in the Discussion of the manuscript to highlight this important question that the reviewer has raised.

• Discussion and Limitations:

1. The discussion section could benefit from an explicit discussion of the study's limitations. What limitations or potential sources of bias should readers be aware of when interpreting the results? 

Response: We are grateful to the reviewers for noting the insufficiency in the discussion section and the valuable suggestions. Our study has several notable limitations. First, most of the findings included in this study are supported by association-based approaches, and further research needs to be conducted to investigate the functional role of endolysosomes in TAM polarization and maintenance. Second, for our gene expression correlation analyses, it is of critical importance to consider the confounding variables that could affect the accurate interpretation of the correlation results, such as the different GBM subtypes (e.g., proneural, classical, mesenchymal), genders, and the heterogeneous genetic background of GBM tumors. Accordingly, we have included a discussion clarifying this point in the revised manuscript (in the subtitle ‘Limitations” in the Discussion section).

2. How do the findings of this study align with or differ from previous research on lysosomes and their role in cancer, particularly in GBM?

Response: We appreciate the reviewer for sharing these pivotal discussion points, allowing us to extend the scope of this manuscript and present our findings in a broader context. In this study, we demonstrated the association between the endolysosomal system and the immunosuppressive TAMs in GBM. In support, previous literature has provided ample evidence showing that lysosomal deregulation is tightly coupled with macrophage polarization, and pharmacological disruption of lysosomal function (e.g., chloroquine) drives TAMs towards more immunosuppressive phenotype (9). Critical endolysosomal genes, such as GRN and FUCA1, have also been shown to contribute to lysosomal dysregulation and neuroinflammation in neurodegenerative diseases, which are known to feature elevated microglia activation. Moreover, aligning with our findings, these two genes have been directly linked to immunosuppression and macrophage infiltration, respectively, in independent studies. Accordingly, we have included a discussion clarifying this point in the revised manuscript.

• Organization and Presentation:

1. Would you consider adding more structured subheadings to the discussion section to enhance the organization of your arguments and conclusions?

Response: We express our gratitude to the reviewers for the valuable feedback that contributed to the improvement of this manuscript’s overall quality. Accordingly, we have added subheadings to the discussion section and restructured the paragraphs to enhance the organization of our arguments and conclusions.

2. Could you provide more context for the figures and tables in the article? Some readers may benefit from additional explanations in the figure captions.

Response: We thank the reviewers for the constructive suggestions that enhanced the quality of this manuscript. Accordingly, we have included additional descriptions in the caption of each figure and table to provide more comprehensive explanations to the readers.

Reference

1. Racle J, Gfeller D. EPIC: A Tool to Estimate the Proportions of Different Cell Types from Bulk Gene Expression Data. Methods Mol Biol. 2020;2120:233-48.

2. Finotello F, Mayer C, Plattner C, Laschober G, Rieder D, Hackl H, et al. Molecular and pharmacological modulators of the tumor immune contexture revealed by deconvolution of RNA-seq data. Genome Med. 2019;11(1):34.

3. Hansen LJ, Yang R, Roso K, Wang W, Chen L, Yang Q, et al. MTAP loss correlates with an immunosuppressive profile in GBM and its substrate MTA stimulates alternative macrophage polarization. Scientific Reports. 2022;12(1):4183.

4. Andersen JK, Miletic H, Hossain JA. Tumor-Associated Macrophages in Gliomas-Basic Insights and Treatment Opportunities. Cancers (Basel). 2022;14(5).

5. Wang G, Zhong K, Wang Z, Zhang Z, Tang X, Tong A, et al. Tumor-associated microglia and macrophages in glioblastoma: From basic insights to therapeutic opportunities. Frontiers in Immunology. 2022;13.

6. Huang SC-C, Everts B, Ivanova Y, O'Sullivan D, Nascimento M, Smith AM, et al. Cell-intrinsic lysosomal lipolysis is essential for alternative activation of macrophages. Nature Immunology. 2014;15(9):846-55.

7. Kimura T, Nada S, Takegahara N, Okuno T, Nojima S, Kang S, et al. Polarization of M2 macrophages requires Lamtor1 that integrates cytokine and amino-acid signals. Nature Communications. 2016;7(1):13130.

8. Tang M, Chen B, Xia H, Pan M, Zhao R, Zhou J, et al. pH-gated nanoparticles selectively regulate lysosomal function of tumour-associated macrophages for cancer immunotherapy. Nature Communications. 2023;14(1):5888.

9. Chen D, Xie J, Fiskesund R, Dong W, Liang X, Lv J, et al. Chloroquine modulates antitumor immune response by resetting tumor-associated macrophages toward M1 phenotype. Nat Commun. 2018;9(1):873.

---

## [Decision Letter · Decision Letter 1]

16 Feb 2024

Gene expression analysis suggests immunosuppressive roles of endolysosomes in glioblastoma

PONE-D-23-33821R1

Dear Dr. He,

We’re pleased to inform you that your manuscript has been judged scientifically suitable for publication and will be formally accepted for publication once it meets all outstanding technical requirements.

Kind regards,

Syed M. Faisal, Ph.D.

Academic Editor

PLOS ONE

Additional Editor Comments (optional):

Reviewers' comments:

Reviewer's Responses to Questions

**Comments to the Author**

1. If the authors have adequately addressed your comments raised in a previous round of review and you feel that this manuscript is now acceptable for publication, you may indicate that here to bypass the “Comments to the Author” section, enter your conflict of interest statement in the “Confidential to Editor” section, and submit your "Accept" recommendation.

Reviewer #1: All comments have been addressed

2. Is the manuscript technically sound, and do the data support the conclusions?

Reviewer #1: Yes

3. Has the statistical analysis been performed appropriately and rigorously? 

Reviewer #1: Yes

4. Have the authors made all data underlying the findings in their manuscript fully available?

Reviewer #1: Yes

5. Is the manuscript presented in an intelligible fashion and written in standard English?

Reviewer #1: Yes

6. Review Comments to the Author

Reviewer #1: The authors have addressed all the comments and therefore the manuscript can be accepted in the present form

7. PLOS authors have the option to publish the peer review history of their article (what does this mean?). If published, this will include your full peer review and any attached files.

Reviewer #1: **Yes: **Sidra Islam

---

## [Editor Report · Acceptance letter]

11 Mar 2024

PONE-D-23-33821R1 

PLOS ONE

Dear Dr. He, 

I'm pleased to inform you that your manuscript has been deemed suitable for publication in PLOS ONE. Congratulations! Your manuscript is now being handed over to our production team.

Kind regards, 

on behalf of

Dr. Syed M. Faisal 

Academic Editor

PLOS ONE